# A Systematic Review of the Evidence of Hematopoietic Stem Cell Differentiation to Fibroblasts

**DOI:** 10.3390/biomedicines10123063

**Published:** 2022-11-28

**Authors:** Bernard J. Smilde, Esmée Botman, Teun J. de Vries, Ralph de Vries, Dimitra Micha, Ton Schoenmaker, Jeroen J. W. M. Janssen, Elisabeth M. W. Eekhoff

**Affiliations:** 1Department of Internal Medicine Section Endocrinology, Amsterdam UMC Location Vrije Universiteit Amsterdam, 1081 HV Amsterdam, The Netherlands; 2Amsterdam Movement Sciences, 1081 HV Amsterdam, The Netherlands; 3Department of Periodontology, Academic Centre for Dentistry Amsterdam (ACTA), University of Amsterdam and VU University, 1081 LA Amsterdam, The Netherlands; 4Medical Library, Amsterdam UMC Location Vrije Universiteit Amsterdam, 1081 HV Amsterdam, The Netherlands; 5Department of Human Genetics, Amsterdam University Medical Center, 1081 HV Amsterdam, The Netherlands; 6Department of Haematology, Radboud UMC, 6525 GA Nijmegen, The Netherlands

**Keywords:** fibroblast, hematopoietic stem cell transplantation, lineage, mesenchymal stem cell, origin, fibrosis

## Abstract

Fibroblasts have an important role in the maintenance of the extracellular matrix of connective tissues by producing and remodelling extracellular matrix proteins. They are indispensable for physiological processes, and as such also associate with many pathological conditions. In recent years, a number of studies have identified donor-derived fibroblasts in various tissues of bone marrow transplant recipients, while others could not replicate these findings. In this systematic review, we provide an overview of the current literature regarding the differentiation of hematopoietic stem cells into fibroblasts in various tissues. PubMed, Embase, and Web of Science (Core Collection) were systematically searched for original articles concerning fibroblast origin after hematopoietic stem cell transplantation in collaboration with a medical information specialist. Our search found 5421 studies, of which 151 were analysed for full-text analysis by two authors independently, resulting in the inclusion of 104 studies. Only studies in animals and humans, in which at least one marker was used for fibroblast identification, were included. The results were described per organ of fibroblast engraftment. We show that nearly all mouse and human organs show evidence of fibroblasts of hematopoietic stem cell transfer origin. Despite significant heterogeneity in the included studies, most demonstrate a significant presence of fibroblasts of hematopoietic lineage in non-hematopoietic tissues. This presence appears to increase after the occurrence of tissue damage.

## 1. Introduction

Fibroblasts are the predominant collagen-producing cells. They have an important structural role in extracellular matrix (ECM) deposition and remodelling and are essential for vital biological processes such as tissue repair and maintenance. Furthermore, important immunological and angiogenic functions through the paracrine excretion of cytokines and growth factors have been also attributed to fibroblasts [1]. Contrastingly, they are also crucial for several pathological processes, such as organ fibrosis, and play a key role in many diseases including dystrophic epidermolysis bullosa, leptin deficiency, and even rare bone diseases such as osteogenesis imperfecta [2]. Despite their importance, the origin of fibroblasts has been a subject of intense debate.

Although it has been long assumed that local mesenchymal precursor cells give rise to fibroblasts, several studies have indicated a possible (partial) bone marrow origin. Following allogeneic hematopoietic stem cell transplant (HSCT), donor-derived cells have been found in non-hematopoietic tissues such as skin, liver, and myocardium [3,4,5]. This may seem remarkable since hematopoietic cells are in principle of a different lineage than the mesenchymal pool. Evidence for such differentiation has been shown through several methods of lineage tracing. In humans, these cells were identified by Y-chromosome detection after male-to-female sex-mismatch transplantations or short tandem repeats chimerism assays after the culturing of fibroblasts from an organ biopsy after transplantation. In animal models, a Green Fluorescent Protein (GFP)-labelled hematopoietic transplant is often used, enabling the visualisation of cells of donor origin. However, other studies using similar methods reported no or minimal contribution of the hematopoietic system to the local fibroblast population in human and animal models, indicating that the possible hematopoietic contribution of fibroblasts in receptor organs is still controversial [6,7].

As knowledge of the origin of fibroblasts after HSCT might open up possible treatment strategies for non-hematological diseases, it is important to have a clear overview of the body of evidence supporting this cell-type transition. This study offers a comprehensive literature review of the evidence supporting the differentiation of hematopoietic stem cells to fibroblasts in various tissues following HSCT, aiming to shed light on their origin and biological processes mediating their emergence in different anatomical locations.

## 2. Methods

### 2.1. Search Strategy

A literature search was performed based on the Preferred Reporting Items for Systematic Reviews and Meta-Analyses (PRISMA)-statement [8]. This study was registered in the PROSPERO registry for systematic reviews (CRD42022355237).

To identify all relevant publications, we conducted systematic searches in the bibliographic databases PubMed, Embase, and Web of Science (Core Collection) from inception up to 28 September 2022 in collaboration with a medical information specialist. The following terms were used (including synonyms and closely related words) as index terms or free-text words: “Fibroblast”, “Myofibroblast”, “Hematopoietic”, “Genetic heterogeneity”, and “Cellular origin”.

The references of the identified articles were searched for relevant publications. Duplicate articles were excluded. All languages were accepted. The complete search strategies for all databases can be found in Appendix A.

### 2.2. Selection Process

Two reviewers (BS and EE) independently screened all potentially relevant titles and abstracts for eligibility. If necessary, the full-text article was checked for the eligibility criteria. Differences in judgement were resolved through a consensus procedure. Studies were included if they met the following criteria: (1) published in English; (2) evidence for fibroblasts or derived cells that extends beyond light microscopy cell morphology, indicated by immunohistochemistry (Vimentin, Collagen I, Fibroblast-specific Protein 1 (FSP1), Na/K/ATPase, calponin, platelet-derived growth factor receptor alpha (PDGFRα), alpha Smooth Muscle Actin (αSMA), desmin, keratocan/lumicam or a combination thereof, cell culture, or electron microscopy) was provided; (3) containing patients or animals that have undergone hematopoietic stem cell transplantation; (4) performed in mammals; (5) published as an original article; (6) full-text availability; (7) all types of study design. We excluded studies if they were: (1) published before 1968; (2) studies in solid tumours; (3) certain publication types: editorials, letters, or conference abstracts.

### 2.3. Data Assessment

The full text of the selected articles was obtained for further review. Two reviewers (BS and EE) independently evaluated the methodological quality of the full-text papers using the Study Quality Assessment Tool created by NHLBI for human studies and the SYRCLE’s risk of bias tool for animal studies [9,10]. The results of this quality assessment can be found in Appendix A.

## 3. Results

### 3.1. Search and Selection of Results

The literature search generated a total of 8478 references: 2585 in PubMed, 3197 in Embase, and 2696 in Web of Science. After removing duplicates of references that were selected from more than one database, 5421 references remained. After subsequent screening rounds of titles and abstracts based on the selection criteria described above, 151 remained for full-text analysis. This analysis excluded a further 47 articles. The flow chart of the search and selection process is presented in Figure 1. Articles covering multiple species (*n* = 2) were discussed in both the animal and human sections of the results. Articles covering multiple tissues (*n* = 5) were analysed in all subsections of the tissues that they examine. Figure 2 summarises the tissue types studied and whether or not articles describe evidence for bone marrow-derived fibroblasts in that tissue. All extracted data from all included studies can be found in Appendix A.

### 3.2. Quality Assessment

The majority of articles (82/104) described studies performed with animals (mice or rats). Using the SYRCLE’s risk of bias tool for animal studies these were analysed for quality. The articles describing results in humans were assessed with the National Heart, Lung, and Blood Institute (NHLBI) Study Assessment Tool. Ten studies were determined to be of good quality (9.6%), 94 studies were of fair quality (90.4%) and none of the studies found were considered to be of poor quality. The rating for each study can be found in Appendix A. No articles were excluded based on their quality rating.

### 3.3. Lineage Tracing and Fibroblast Identification Methods

In human studies, the most prevalent form of lineage tracing was based on the use of a sex mismatch transplant with the detection of the Y-chromosome in the studied tissue (22/24). These studies identified transplant origin in organs making use of in situ hybridisation. Other studies in humans investigated cell tracing with the detection of variation in short tandem repeats, which make use of unique, tandemly repeated DNA motifs that differ between host and donor.

In animals, a hematopoietic stem cell (HSC) graft from Green Fluorescent Protein (GFP)-labelled donor animals was most frequently used, either solely (53/82) or in combination with another lineage tracking method such as sex-mismatch or genetic knockout (12/82). Other methods included sex-mismatch transplants or specific gene-knockout transplants.

Most articles (49/104) used immunohistochemistry with a general fibroblast marker (Vimentin, Collagen I, FSP1, Na/K/ATPase, calponin, PDGFRα) or a combination thereof. Other articles used immunohistochemical markers for more specialised or tissue-specific fibroblasts: alpha Smooth Muscle Actin (αSMA) for myofibroblasts (36/104); desmin for pancreatic stellate cells (1/104); keratocan/lumicam for corneal keratocytes. Finally, fibroblast cultures (13/104) and electron microscopy (2/104) were used as methods of fibroblast identification.

Below, we first summarise all the evidence about hematopoietic cell transplantation-derived fibroblast populations in the various organs derived from animal studies, followed by evidence obtained in humans.

### 3.4. Animals

#### 3.4.1. Adipose Tissue

Fibroblasts in adipose tissue after GFP-labelled bone marrow transplantation (BMT) were evaluated in mice by Guerrero-Juarez et al. [11]. This study found that bone marrow-derived cells (BMDCs) had differentiated into fibroblasts in adipose tissue as well as into adipocytes after purified hematopoietic stem cell (HST) transplantation.

#### 3.4.2. Adrenal Gland

Direkze et al. described the presence of donor-derived α-SMA-positive cells, indicating myofibroblasts, in the adrenal capsule after sex-mismatch BMT in mice [5]. The number of donor-derived α-SMA-positive cells was not quantified.

#### 3.4.3. Arterial Wall

Four studies were found describing the origin of arterial fibroblasts in mice after HSCT [7,12,13,14]. All studies confirmed the presence of BMDCs within the arterial wall after BMT to varying degrees. Campbell et al. [15], Zou et al. [14], and Rodriguez-Menocal et al. [13] used a model with intimal damage (scratch or balloon) and found that 2, 43–56% of α-SMA-positive cells in the neointima were donor-derived. Zieggelhoeffer et al. [7], using a model of femoral collateral artery occlusion, identified donor-derived fibroblasts, based on the colocalisation of GFP and vimentin, indicative of the fibroblast lineage. Cells of donor origin were further found to contribute to pericytes and leukocytes but not to endothelial cells.

#### 3.4.4. Bladder

Two studies described the analyses of bladder tissue after GFP-labelled BMT [16,17]. Both studies demonstrated the presence of GFP-positive cells in the bladder wall after transplantation. Double positivity for GFP and α-SMA indicated that these cells were myofibroblasts, while double positivity for GFP and AE1/AE3 shows that bone marrow-derived cells contributed to the specialised epithelium of the bladder, the urothelium.

#### 3.4.5. Bone, Cartilage and Synovium

Our search found three articles describing the involvement of BMDCs in bone [18,19,20]. Pereira et al. [19] described donor-derived collagen type I-producing fibroblasts in primary cultures from the long bones after BMT. Bilic-Curcic et al. [18] found that heterotopic bone, formed in a subcutaneous collagen gel implant, contained both fibroblasts (marker) and osteoblasts (marker) of donor origin. In another model of heterotopic ossification, Prados et al. [20] were not able to show such contributions of donor cells.

In fibroblast cultures from cartilage, Pereira et al. [19] showed that 8% of fibroblasts were donor-derived after BMT.

One study focused on the contribution of BMDCs to synovial repair [21]. This study found a small number of donor-derived fibroblasts in the synovium after joint surface injury.

#### 3.4.6. Bone Marrow Stroma

Ch’ang et al. [22], Ebihara et al. [23], Gorskaya et al. [24], and Yokota et al. [25] analysed the contribution of donor bone marrow stromal cells in murine models after BMT. Only Ebihara et al. [23] found significant donor contribution to the bone marrow stroma, evidenced by the formation of the BMT-derived fibroblast colony-forming units (CFU-F). CFU-F colonies were also studied by Gorskaya et al. [24] but were not shown to be donor-derived.

#### 3.4.7. Eye

Murine studies were identified involving two parts of the eye: five cornea studies and one sclera study [26,27,28,29,30,31]. All studies used GFP-labelled transplants, and all found a contribution of BMDCs. Corneal myofibroblasts (αSMA+), pericytes (Ng2 proteoglycan+), and lymphatic vessels (VEGRF2+) all had a donor-derived component. Keratocytes are specialised fibroblasts of the cornea. Harada et al. [27] also found donor-derived keratocytes, but this was not seen by Liu et al. [29].

BMDCs contributed to scleral fibroblasts and antigen-presenting cell populations [28]. This was more pronounced after inflammation.

#### 3.4.8. Gastrointestinal and Peritoneum

Bone marrow-derived cells contributing to the (myo)fibroblast population in the stomach, small intestine, and colon of rats and mice were shown in all seven included studies [5,32,33,34,35,36,37]. Lee et al. [37] observed pericryptal engraftment in the stomach but no epithelial engraftment. Several models of inflammation/damage induction reported an increase in engrafted (myo)fibroblasts in gastrointestinal tissues. Epithelial cells appeared to be donor-derived to a smaller degree.

Campbell et al. [15] implanted silastic tubing or boiled blood clots in the peritoneal cavity and determined that most myofibroblasts were donor-derived (Y-chromosome +) 14 days after implantation [15].

#### 3.4.9. Heart and Heart Valves

Of seven articles describing the origin of fibroblasts and/or cardiomyocytes in the heart, two did not find donor contribution to the fibroblast population and two did not find a contribution to cardiomyocytes [6,38,39,40,41,42,43]. All studies used a form of cardiac damage, such as infarction or parasitic infection. In the studies confirming BMDC contribution to cardiomyocytes, donor-derived cells were scarce. Endo et al. [39] showed less than 0.01% BM-derived cardiomyocytes, whereas Mölmann et al. [40] were only able to find three cardiomyocytes (titin-positive) in 25 examined hearts. Van Amerongen et al. [43] observed 24% BM-derived myofibroblasts in an infarcted area.

Both articles describing the origin of cells in the cardiac valves showed BMDC contribution [44,45]. Viscoti et al. [45] used a single-cell HSC transplant. Hajdu et al. [44] showed the contribution of these cells to collagen production in cardiac valves.

#### 3.4.10. Inner Ear

Lang et al. [46] performed a single-cell HSC transplant, in which clonally expanded HSCs were transplanted in mice and concluded that CD45-negative mesenchymal cells were partially BM-derived, including fibrocytes.

#### 3.4.11. Kidney

Fibroblasts in kidney tissue were studied in six papers [5,25,47,48,49,50]. Five of them showed donor-derived (myo)fibroblasts in the kidney. The percentage of donor-derived (myo)fibroblasts as a total of the (myo)fibroblasts present varied from 8.6% to 90% (after urethral obstruction). Li et al. [49] further showed that there were donor-derived endothelial cells (von Willebrand factor (vWF)-positive) in the kidney. In contrast, Yokota et al. [25] reported no presence of GFP-labelled α-SMA-positive mesangial cells (myofibroblasts) after BMT in a model without kidney injury.

#### 3.4.12. Liver and Bile Ducts

Eight studies were found describing the fate of BMDCs in damaged hepatic tissue [51,52,53,54,55,56,57,58]. Of these eight studies, seven were able to show the presence of BMDC-derived fibroblasts. Only in the study by Kisselava et al. [56], no CD45-negative (non-hematopoietic) donor-derived cells were found. The seven studies that described the presence of donor-derived (myo)fibroblasts estimated their presence between 10 and 80% of the total fibroblast population within fibrotic lesions. In most cases, a model of liver fibrosis was used, which increased the amount of donor-derived (myo)fibroblasts in the liver. García et al. [53] also reported a small percentage of donor-derived hepatocytes (albumin positive).

Roderfeld et al. [59] used a model of sclerosing cholangitis and found that both periductular myofibroblasts in bile ducts and fibrocytes were donor-derived without quantifying their relative contribution.

#### 3.4.13. Lung

Our search found thirteen mouse and rat studies reporting on the pulmonary system [5,60,61,62,63,64,65,66,67,68,69,70,71]. In all thirteen studies, there was evidence of bone marrow-derived fibroblasts in pulmonary tissue after BMT. Most studies made use of various forms of fibrosis induction, such as bleomycin treatment, which increased the number of donor-derived (myo)fibroblasts up to 41% of the total population. McDonald et al. [66] confirmed these results by clonally expanding single-cell BMT. Spees et al. [71] reported donor-derived pulmonary epithelial cells and vascular epithelial cells. Donor-derived vascular epithelial cells, as shown by vWF or CD31 positivity, in the lung were also found by Angelini et al. [60], whereas Serikov et al. [69] failed to identify epithelial or endothelial cells of donor origin in the lung.

#### 3.4.14. Pancreas

In the pancreas, BMDCs were found to contribute to fibroblast and pancreatic stellate cell populations in all five studies found by our search [72,73,74,75,76]. Akita et al. [72], Lin et al. [73], and Marrache et al. [74] found that the number of donor-derived fibroblasts and pancreatic stellate cells, a specialised form of pancreatic fibroblasts which expresses desmin, increased when a pancreatitis/pancreatic fibrosis model was used. In the study by Akita et al. [72] up to 23.3% of the pancreatic stellate cells were donor-derived, which peaked at one week after fibrosis induction.

#### 3.4.15. Skin

The contribution of BMDCs to the skin tissue of mice was investigated in twelve studies [5,11,65,77,78,79,80,81,82,83,84,85]. Eleven of these studies demonstrated donor-derived (myo)fibroblasts in the skin, with the exception of Boban et al. [79]. All articles utilised a form of skin wounding or fibrosis leading to increased donor-derived fibroblasts after the injury. Rea et al. [84] found that BMDCs persisted for 120 days after the injury was inflicted.

#### 3.4.16. Teeth and Palate

In two studies using GFP-labelled transplants for tracing BMDC contribution to dental tissues, BMDCs were found to differentiate into several cell types [86,87]. Both showed that fibroblasts in dental tissues can be bone marrow-derived. Wilson et al. [87] used the transplantation of a clonal population derived from a single HSC (single cell transplantation) to prove the hematopoietic origin of the donor-derived fibroblasts and showed that they produced collagen.

Both studies examining the palate by Verstappen et al. [85,88] showed a donor-derived component of (activated) fibroblasts after GFP-labelled BMT that increased following wounding.

### 3.5. Human Studies

#### 3.5.1. Arterial Wall

Kvaniska et al. [89] studied bone-marrow arteries in nine patients 9 to 1172 days after sex-mismatch BMT. Myofibroblasts (αSMA positive) were found to be exclusively of host origin, whereas up to 30% of BM vascular endothelial cells (CD34 positive cells lining arteries) were host-derived, with a peak at three months after BMT.

#### 3.5.2. Bone Marrow Stroma

Eleven human studies describe the lineage of bone marrow stroma after BMT [90,91,92,93,94,95,96,97,98,99,100]. Ten out of these studies found no contribution from donor-derived cells to bone marrow stroma or cultures thereof, using both sex-mismatch and short tandem repeat methods for lineage tracing. Cilloni et al. [92] reported a limited capacity for donor-derived cells for the reconstitution of the bone marrow stroma. In bone marrow stroma cultures taken from 14 patients in complete hematological remission 1–79 months after receiving sex-mismatched T-cell depleted allografts, two patients showed mixed chimerism.

#### 3.5.3. Eye

Three studies were found describing BMDCs in the conjunctiva of humans with ocular grafts versus host disease [101,102,103]. All three employed sex-mismatch lineage tracing and found that BMDCs contributed to (myo)fibroblasts in the conjunctiva.

Eberwein et al. [101] studied conjunctival tissue specimens from seven patients 9 to 40 months after BMT. Six out of these seven patients had active graft versus host disease. Donor-derived endothelial cells and myofibroblasts were found to constitute around 5.4% of all cells with no clear relation with time since transplantation. Hallberg et al. performed two studies for which the first [103] 17 samples from nine patients were collected, whereas the second [102] included 70 samples from 46 patients. In both studies, approximately 9% of myofibroblasts were determined to be of donor origin, which increased gradually over time.

#### 3.5.4. Lacrimal Gland

Using lacrimal gland biopsies taken from seven patients suffering from dry eye 5–36 months after sex mismatch BMT, one study found that 13.4% to 26.7% of fibroblasts in the fibrotic lesion were donor-derived [104]. In this study, donor-derived cells were also found in the acini and ductal epithelium, but the phenotype of these cells was not analysed. No relation was found between the severity of the dry eye conditions and the percentage of donor-derived fibroblasts. The relationship with time since transplant was not analysed.

#### 3.5.5. Liver and Bile Ducts

Forbes et al. [105] used a dual approach to investigate the origin of (myo)fibroblasts in (fibrotic) liver. They studied biopsies of fibrotic lesions in seven sex-mismatch liver transplant recipients and one sex-mismatch BMT recipient who developed hepatitis C cirrhosis. In the liver transplant recipients, 6.8–22.2% of the α-SMA–positive myofibroblasts were donor-derived for 2–8.1 years after transplantation. In the BMT recipients, 12.4% of the myofibroblasts in the fibrotic lesion were donor-derived 10 years after transplantation.

#### 3.5.6. Lung

A dual approach was also used by Bröcker et al. [106] who examined fibrotic lesions in twelve lung transplant recipients and two BMT recipients. In the lung transplant recipients, a mean of 32% of fibroblasts was found to be donor-derived from 5 weeks to 6 years after transplantation. In the BMT recipients, donor-derived fibroblasts in fibrotic lesions 15–36 months after transplantation were detected, but not quantified.

#### 3.5.7. Skin

We identified three studies that describe the fate of BMDCs in the human skin [3,107,108]. Wagner et al. [108] studied six patients with recessive dystrophic epidermolysis bullosa that were treated with allogenic BMT; a median of 20% skin chimerism, persistent up to 198 days after transplantation, was observed. In one patient, it was confirmed that a proportion of these cells were CD45 negative, showing these were not hematopoietic cells.

The other two studies both describe a single patient after sex-mismatch BMT with a dermal graft versus host disease. Goussetis et al. [107] found that a significant number of skin (myo)fibroblasts were donor-derived, but this was not the case in the study of Stewart et al. [3] with dermal fibroblast cultures.

## 4. Discussions

This systematic review investigated to what extent cells from a hematopoietic cell transfusion differentiate into fibroblasts. We present the results of this systematic review based on categorisation by the investigated tissue types and species. Although there is significant heterogeneity in the results reported by various studies, there is clear evidence pointing to the presence of a small but non-negligible number of fibroblasts of hematopoietic lineage in non-hematopoietic tissues. This number of cells seems to increase when tissue damage occurs.

In humans, BM transplant-derived fibroblasts can be found in arteries, conjunctiva, lungs, heart valves, conjunctiva, the liver, the skin, and the lacrimal gland. It has been also consistently shown that bone marrow stroma remains of host origin after hematopoietic stem cell transplantation. Although the functional contribution of BM-derived cells to the respective tissues still needs to be enlightened, it is noteworthy that the prevalence of these cells increases in damaged or inflamed tissues even when correcting for the presence of CD45-positive leukocytes. In fibrotic lesions in the liver, for instance, high percentages of donor-derived fibroblasts were found in both experimental mouse (10–80%) [51,52,53,54,55,56,57,58] and human studies (6.8–22.2%) [105]. Due to practical and ethical constraints in obtaining multiple tissues at different time points, there is a paucity of data in humans regarding the time and method of the engraftment of BM-derived cells into the various tissues. In our current review, the maximum time between engraftment and the ability to detect fibroblasts in the recipient tissue is 122 months, suggesting that grafted cells may survive and contribute to homeostasis or disease over a long period.

In mice, additional evidence exists for the presence of BM-derived fibroblasts in many other tissues, such as the urinary bladder, arteries, and bone. Indeed, most tissues physiologically contain some form of BM-derived fibroblast population, which is greatly increased after injury.

Some authors have suggested that HSCs do not differentiate into fibroblasts or other cell types, but rather that the appearance of donor-derived cells is caused by the fusion of marrow-derived cells with resident tissue cells. This ability of HSCs has been shown to occur in, for example, hepatocytes, cardiomyocytes, and neurons [109,110]. Multiple studies explicitly looked into this possibility by searching for polykarions, but none found evidence that this occurred to any substantial degree [23,39,46,70].

When a bone marrow transplant is infused, stroma cells and mesenchymal stem cells (MSCs) are transplanted along with HSCs. One could argue that bone marrow-derived fibroblasts could originate from these cells instead of HSCs. However, donor-derived fibroblasts have been found in tissues several years after BMT, far longer than the life span of a fibroblast, while there is consensus that the bone marrow stroma, containing MSCs, of BMT recipients remains of donor origin [90,91,92,93,94,95,96,97,98,99,100]. This suggests that, unless fibroblasts from the donor contain mesenchymal stem cell properties that can replenish the local fibroblast pool once migrated into the tissue, these cells are replenished from the hematopoietic source. Furthermore, single-cell HSC transplantation, in which a number of clones from single hematopoietic stem cells are transplanted without other cell types, and purified HSC transplantations were performed in multiple studies [11,23,39,45,46,66,87,111]. These studies found no substantial differences in the bone marrow-derived fibroblasts even after the use of much purer transplantations compared to the transplantation of whole bone marrow across a variety of tissues.

Taken together, the current literature suggests that a subtype of hematopoietic cells is capable of transdifferentiation into (myo)fibroblasts in most tissues with varying degrees of contribution to functional properties such as collagen production. Many studies have failed to evaluate long-term BM contribution. The few studies that addressed this issue, generally stated that the BM-derived (myo)fibroblast population peaks several weeks after a form of injury which subsequently recedes [11,36,53,69,84]. This implicates the presence of bone marrow-derived cells in non-hematopoietic tissues in which they primarily serve as supporting cells in repairing tissues, rather than sustaining long-term engraftment. They, therefore, appear to follow the same pattern of inflammatory and proliferative phases in physiological wound healing [112].

Apart from the transplant evidence, one could speculate about whether the process of renewal by hematopoietic stem cells could also be a naturally occurring phenomenon. The ability of hematopoietic stem cells to differentiate into (myo)fibroblasts and other cell types may have significant implications for a range of pathophysiological processes. Examples include organ fibrosis, dystrophic epidermolysis bullosa, leptin deficiency and even rare bone diseases such as fibrodysplasia ossificans progressiva in which the process of heterotopic ossification is possibly facilitated through fibroblasts [1,113,114,115,116,117]. It is plausible that the properties of HSCs may potentially affect disease presentation. However, much is still unknown about the heterogeneity of donor-derived fibroblasts and the conditions that enable HSCs to transdifferentiate. Furthermore, only a minority of all fibroblasts appear to be BM-derived, possibly diminishing the clinical impact of HSCT on fibroblast-related disease. Future research exploring these unknown areas is needed to find practical applications for BMT in non-hematological diseases.

## Figures and Tables

**Figure 1 biomedicines-10-03063-f001:**
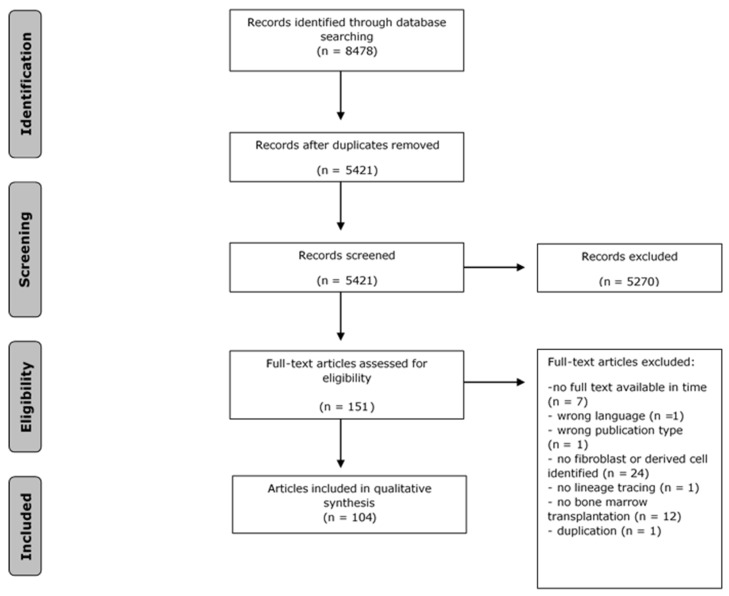
Flow chart of the literature search and selection process.

**Figure 2 biomedicines-10-03063-f002:**
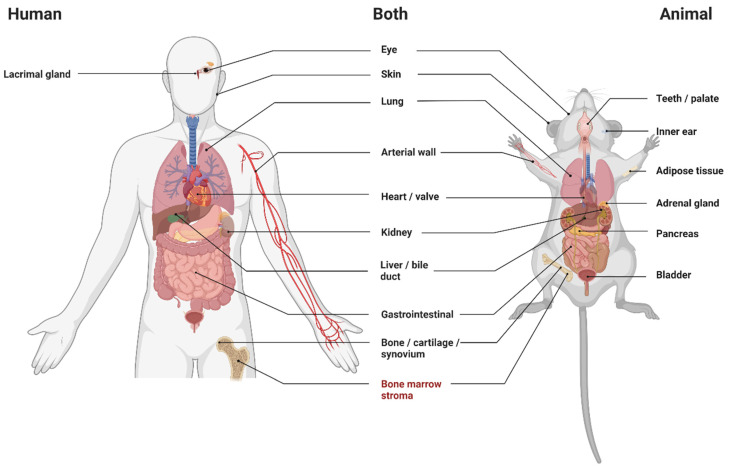
Tissues studied in included articles, by species. Black text = studies identified bone marrow-derived fibroblasts. Red text = studies did not identify bone marrow-derived fibroblasts. Created with BioRender.com.

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
