# Peer review of "A Systematic Review of the Evidence of Hematopoietic Stem Cell Differentiation to Fibroblasts"

_biomedicines, 2022, doi:10.3390/biomedicines10123063_

Round 1

Reviewer 1 Report

In this systematic review the authors provide an overview of the current literature regarding the differentiation of hematopoietic stem cells into fibroblasts in various tissues. They showed the presence of a small but non-negligible number of fibroblasts of hematopoietic lineage in non-hematopoietic tissues. This comprehensive review is informative.

I have some comments.

1.     This number of cells seems to increase when tissue damage occurs. What is the mechanism of this phenomenon?

2.     There are no reports of nervous system. Is there any possibility of the differentiation of hematopoietic stem cells into nervous system tissues?

Reviewer 2 Report

The manuscript is well written, interesting and suitable for publication. I have some minor remarks described below.

The title is suitable, perhaps the author s may add at the end ‘a short review’. This is up to the authors, we only suggest this, as the paper is not very long: A systematic review on the evidence of hematopoietic stem cell differentiation to fibroblasts: s short review.

The Introduction is well written and describes the relevant topic.

Methods: It is interesting that there were not many articles analysed from the whole collection. It is therefore a very specific topic.

The Discussion is well written.

My remarks:

In the Results, the authors state: ‘After subsequent screening rounds of titles and abstracts, 151 remained for full-text analysis.’ On what basis did the authors exclude these articles and choose the relevant one? Can they describe?

Round 2

Reviewer 1 Report

The authors responded to my concerns.